# Regional, Ideological and Inheritable Characteristics of Knowledge: A Survey of Three Compilations of Buddhist Encyclopedias in China from 1950s to 2000s

Wenli Fan [1,2]

1    School of Philosophy, University of Chinese Academy of Social Sciences, Beijing 100732, China; wenlifan@outlook.com

2    Institute of Philosophy, Chinese Academy of Social Sciences, Beijing 100732, China

**Abstract:** Three official compilations of Buddhist encyclopedias were undertaken in China between the 1950s and 2000s. A sociological examination of these compilations reveals notable characteristics of the Buddhist knowledge system. Firstly, the production of knowledge manifests distinct regional attributes; it is not a process of standardization or objectification, but reflects local idiosyncrasies determined by its place of origination. Secondly, the majority of modern encyclopedia compilations are integral to the construction of national knowledge systems; hence, a nation's ideological tendencies profoundly influence the articulation of knowledge. Lastly, knowledge is transferred through two mediums: texts and people. Given the immutability of classical knowledge and the consistency of knowledge producers during this period, the results of the three compilations exhibit numerous commonalities.

**Keywords:** encyclopedia; Chinese Buddhism; sociology of knowledge; knowledge system

## 1. Introduction

Between the 1950s and 2000s, China embarked on three official compilations of Buddhist encyclopedias: the compilation of the English *Encyclopedia of Chinese Buddhism* commencing in 1956 (abbreviated as ECB1956); the compilation and publication of the five-volume Chinese *Zhongguo Fojiao* 中國佛教 (Chinese Buddhism) beginning in 1979 (abbreviated as ZF1979); and the compilation of Buddhist entries in the *Zhongguo dabaike quanshu: Zongjiao* 中國大百科全書·宗教 (Encyclopedia of China—Religion) starting in 1980 (abbreviated as ZDQZ1980). The three compilations exhibit both commonalities and variances, as presented in the following table (Table 1).

The working files and historical records of these compilations have been preserved, providing insights into the endeavors of the Chinese Buddhist community to establish their Buddhist knowledge system. Previous studies (Fan 2018; Huang 1994; Sodō 1968) focus on the historical details of these events, yet systematic and in-depth analyses are lacking and numerous questions are still to be answered. From the perspective of the social history of knowledge, "The selection, organization and presentation of knowledge is not a neutral, value-free process. On the contrary, it is the expression of a world-view supported by an economic as well as a social and political system." (Burke 2000, p. 176). For instance, in ECB1956, a project carried out in China but obligated to supply information for a Sri Lankan encyclopedia that claimed to present a global vision, how does the regional shift influence the articulation of knowledge? The compilations discussed are all initiated by official departments or national governments, so how does the official background influence the knowledge system created by these encyclopedias? Three compilations of Buddhist encyclopedias were undertaken in China in such a short period—how and why are they similar and different? This paper explores the distinctions and overlaps in the three compilations from three standpoints—the regional, ideological, and inheritable characteristics

of knowledge—with the intention to foster a profound comprehension of the construction of the Buddhist knowledge system in China.

**Table 1.** Three compilations of Buddhist encyclopedias.

| Abbreviated Name | ECB1956 | ZF1979 | ZDQZ1980 |
|---|---|---|---|
| Full Description of the Project | The compilation of the English *Encyclopedia of Chinese Buddhism* commencing in 1956 | The compilation and publication of the five-volume Chinese *Zhongguo Fojiao* 中國佛教 (Chinese Buddhism) beginning in 1979 | The compilation of Buddhist entries in the *Zhongguo dabaike quanshu: Zongjiao* 中國大百科全書·宗教 (Encyclopedia of China—Religion) starting in 1980 |
| Sponsor | Sri Lankan government | BAC | Chinese government |
| Project Executor | Buddhist Association of China (中國佛教協會, abbreviated as BAC) | BAC | BAC and the Institute of South Asian Studies of the Chinese Academy of Social Sciences |
| Editor-in-Chief | BAC[1] | BAC | Ju Zan 巨贊, Huang Xinchuan 黄心川[2] |
| Number of Participants | 38[3] | 59 (38 + 21)[4] | 67[5] |
| Working Period | 1956 to the middle of 1960s | 1979–2004 | 1980–1988 |
| Feature of Independency | Part of a comprehensive Buddhist encyclopedia | An independent encyclopedia in Chinese Buddhism | Part of a religious volume in a Grand Encyclopedia |

## 2. The Three Compilations of Buddhist Encyclopedias

The ECB1956 was triggered by the *Encyclopaedia of Buddhism* project initiated by Sri Lanka. Upon the invitation of Sri Lanka's Prime Minister John Lionel Kotelawala, the Premier of China, Zhou Enlai 周恩來, acceded to China's commitment to compile the section related to Chinese Buddhism. This task was assigned to the Buddhist Association of China. The BAC promptly set up the "Editorial Committee for *Encyclopedia of Chinese Buddhism*" at Guangji Temple 廣濟寺. Zhao Puchu 趙樸初, the Vice President and Secretary-general of BAC, assumed the role of chairman of the Editorial Committee, with renowned Buddhist scholar, Lü Cheng 呂澂, as Vice Chairman. The committee, based at the Jinling Scriptural Press (Jinling kejingchu 金陵刻經處) in Nanjing, enlisted numerous professionals to author entries for this encyclopedia, including Fazun 法尊, Gao Guanru 高觀如, Guo Yuanxing 郭元興, Zhang Keqiang 張克強, Huang Chanhua 黃懺華, Juzan 巨贊, Li An 李安, Lin Ziqing 林子青, Longlian 隆蓮, Mingzhen 明真, Su Jinren 蘇晉仁, Tian Guanglie 田光烈, Yu Yu 虞愚 and Zhou Shujia 周叔迦, among others. The workflow of the compilation involved designing the framework and contents of entries; drafting by authors in Chinese; the review and revision of the Chinese drafts by the committee; the translation of the final Chinese drafts into English by the translation team (See Figure 1); and another round of review and revision of the English drafts by the committee. After nearly a decade of effort,

by 2 February 1966, the committee had generated a total of 357 English manuscripts for the Sri Lankan Buddhist encyclopedia, 154 of which had been sent to Sri Lanka (Fan 2018, p. 130). The dispatched articles were disassembled and integrated into different entries of the Sri Lankan *Encyclopaedia of Buddhism*, dispersed throughout the volumes. The million-word English manuscripts have not been fully published as a single piece yet.[6] Among them, barring a few manuscripts damaged by water, 300 written English manuscripts, totaling 1.15 million words, remain intact. Additionally, the Chinese manuscripts from this compilation were later published in the ZF1979 project.

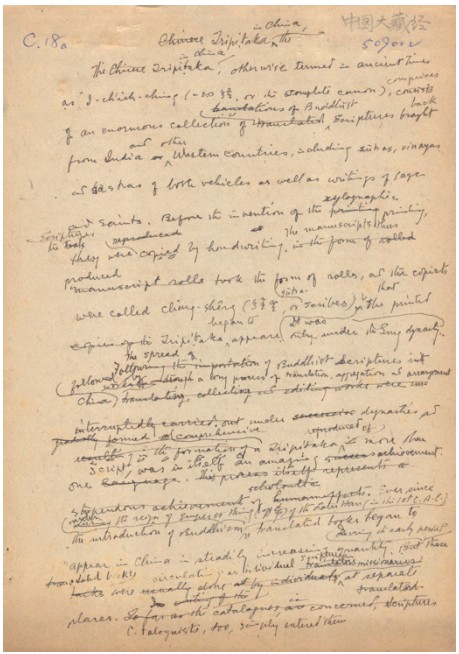

**Figure 1.** An image of the manuscript of *Encyclopedia of Chinese Buddhism* (Entry C18: Chinese Tripiṭaka). Note: The copyright of this picture belongs to Research Institute of Chinese Buddhist Culture of Buddhist Association of China (中國佛教協會中國佛教文化研究所). I have obtained an official usage permission from the institute.

The ZF1979 compilation drew upon the Chinese manuscripts created in the 1950s and 1960s for the ECB1956, with minor additions and modifications. Under the initiative of Zhao Puchu, the then-president of the BAC, and Zheng Guo 正果, the then-leader of the BAC research group, the Chinese manuscripts were organized from 1979 onwards and subsequently published under the title *Zhongguo Fojiao*. In June 1980, the first volume of *Zhongguo Fojiao* was released by Knowledge Publishing House (Zhishi chubanshe 知識出版社) in China. Ming Zhen, Lin Ziqing and Zheng Guo, among others, undertook the principal review work, with Jing Hui 淨慧 and Wang Xin 王新 serving as the primary editors. In the early 1980s, very few newly published Buddhist books existed. This was the first Buddhist encyclopedia compiled by the Chinese Buddhist community following the establishment of the People's Republic of China. The publication garnered a robust social response, with media outlets such as *Guangming Daily* 光明日報 and *Dushu* 讀書 in Beijing, *Wen Hui Po* 文匯報 in Shanghai, *Ta Kung Pao* 大公報 in Hong Kong, and *Hinaka Bukkyō* 日中佛教 (Sino-Japanese Buddhism) in Japan, reporting on the event consecutively (Shuangmu 1981, p. 40). The second volume was published in August 1982, the third and fourth volumes were published in May 1989, and the fifth volume was published in June 2004. Wang Xin primarily oversaw the editing of these subsequent volumes. *Zhongguo Fojiao* largely utilized the older manuscripts composed over 20 years ago, complemented by a small number of new manuscripts. For instance, some articles under the theme of Buddhist teachings in the fourth volume were newly written in the 1980s; and in the fifth volume, about half of the articles on *Tripiṭaka* were newly written.

The compilation of ZDQZ1980 was rooted in the editorial initiative for the *Zhongguo dabaike quanshu* 中國大百科全書 (Encyclopedia of China), which commenced with the "Guanyu bianji chuban *Zhongguo dabaike quanshu* de jianyi" 關於編輯出版《中國大百科全書》的建議 ("Recommendations for Editing and Publishing the Encyclopedia of China"). This initiative was proposed in January 1978 by Jiang Chunfang 姜椿芳, who was then the deputy director of the Central Compilation and Translation Bureau (中央編譯局). Subsequently, several related institutions, including the Chinese Academy of Sciences (中國科學院), the Chinese Academy of Social Sciences (中國社會科學院), and the National Publishing Administration (國家出版事業管理局), collectively signed and presented a "Qingshi baogao"請示報告 ("Report for Instructions") to the Central Committee of the Communist Party of China, proposing the compilation and publication of a *Zhongguo dabaike quanshu*. The Central Committee sanctioned the establishment of the Encyclopedia of China Publishing House (Zhognguo dabaike quanshu chubanshe 中國大百科全書出版社) on 28 May 1978, thereby marking the inception of the *Zhongguo dabaike quanshu* project ([Huang 1994](), pp. 264–65). Among a myriad of subjects, the compilation of the Astronomy Volume was initiated first, and other subjects sequentially followed. The Religion Volume was initiated in 1979 with an initial plan for 420 Buddhist entries, and these were to be compiled by BAC and the Institute of South Asian Studies of the Chinese Academy of Social Sciences (中國社會科學院南亞研究所). To accomplish this task, BAC established the "Buddhist Entries of *Zhongguo dabaike quanshu* Compilation Group" on 20 October 1980, with Juzan at the helm of its organization (*Fayin* [Reporter 1981](), p. 48). The entries were revealed to the world along with the publication of *Zhongguo dabaike quanshu: Zongjiao* in 1988.

## 3. Globality and Locality: Regional Characteristic of Buddhist Knowledge

In Chinese traditional culture, Buddhism holds a unique position as it was not indigenous, but originated from India. Consequently, Chinese Buddhism has exhibited global attributes since its inception. Buddhism began in India, expanded northwest into Central Asia, and subsequently penetrated the inland areas of China, eventually evolving into Han Buddhism. Furthermore, it spread north to the Tibetan region of China and east to Southeast Asia, resulting in Tibetan Buddhism and Theravāda Buddhism in these respective areas. These three Buddhist traditions share similarities, but also hold distinct differences. With the propagation of Buddhism in China, during the Wei, Jin, Southern, and Northern Dynasties, conflicts and confluences occurred between Chinese and Indian cultures. This was followed by an unparalleled flourishing during the Sui and Tang Dynasties, leading to the establishment of a variant of Buddhism, distinct from Indian Buddhism, known as Chinese Buddhism. Subsequent to this period, Chinese Buddhism embarked on a distinct path, with the emergence of numerous local eminent monks and temples. Chinese Buddhists exhibited considerable innovations in ideologies, beliefs, institutions, precepts, and rituals. Beginning from the Tang Dynasty, Chinese Han Buddhism progressively spread to Northeast Asia, spurring new developments in Japan and the Korean Peninsula. Although Buddhism is inherently a global religion, within the Chinese context, it displays potent local characteristics. The narration of Chinese Buddhism, regardless of the author, should invariably focus on elements that underscore its Chinese traits.

The ECB1956 was composed within the framework of the Sri Lankan *Encyclopaedia of Buddhism*. G. P. Malalasekera, the chief editor of the Sri Lankan encyclopedia, noted in the preface of volume 1 that the encyclopedia strives to provide an exhaustive account of Buddhism, inclusive of doctrines, schools and sects, rites and ceremonies, fine arts, shrines and places of pilgrimage, and biographies, among other aspects. He intended to outline the evolution and influences of these facets across diverse countries, regions, and cultural traditions, aspiring to incorporate both Mahāyāna and Theravāda information ([Malalasekera [1997] 1961–1965](), p. 3). However, this ambitious plan was not actualized as anticipated, as is evident even in the trial version of the encyclopedia titled, *Encyclopaedia of Buddhism—Volume of Specimen Articles*, published in 1957. This version aimed

to present the foundational aspects of the encyclopedia and gather feedback ahead of the official volumes.

The trial version garnered attention and critiques from academia. Clay Lancaster noted that "the published articles fall mostly within the scope of the Southern School of Buddhism." (Lancaster 1957, p. 216). Alexander Soper highlighted that within the Volume of Specimen Articles, under the key heading *abhaya* and its suffix compounds, we can find summaries of information about nine historic monks and nuns so named, chiefly Sinhalese; five secular contemporaries of Śākyamuni, a king, a prince and three lesser laymen; three Buddhas and a goddess; the great Anuradhapura monastery Abhayagiri and the sect based thereon; a noted Anuradhapura reservoir-lake; and several Pāli suttas. However, there was no entry for *abhaya mudrā*, a sacred hand gesture or "seal" that is very important in Mahāyāna Buddhism. He considered this as an example of "occasional irregularities or omissions on the Mahāyāna side of the ledger" (Soper 1963, p. 366). Additionally, the editorial board acknowledged that Southern and Northern Buddhism might offer different narratives or perspectives on the same topic. The initial plan for such entries was to incorporate articles from multiple authors, duly signed for completeness. However, J. W. de Jong pointed out that this practice was inconsistently applied. In several instances, topics important to both Theravāda and Mahāyāna were treated almost exclusively from the perspective of Pāli texts, e.g., Abbuda niraya, Abhijhā and Abhiññā (de Jong 1962, p. 381).

Indeed, the global and local aspects of knowledge coexist. As mentioned above, knowledge, a human construct, is not an objective, standardized, and universal entity, but is often constrained by the temporal, spatial, and societal context of its creators. An examination of the process of knowledge production reveals how the subjectivity and limitations of creators profoundly impact the final expression and representation of knowledge.

Consider the entries pertaining to Chinese Buddhism in the Sri Lankan Buddhist encyclopedia: the structural design and entry cataloging were executed by the Sri Lankan editorial board. The Chinese editorial committee supplied first-hand materials through article submissions. The Sri Lankan editorial board then processed and reproduced these materials to align with the overall requirements of the book. Therefore, the final representation of this information was determined by the Sri Lankan editorial board. Sri Lanka aligns with the Theravāda Buddhist tradition. Despite the encyclopedia's proclaimed objective to consider both Theravāda and Mahāyāna Buddhism, the cataloging of specific entries inevitably leaned towards Theravāda Buddhism from the outset. Furthermore, within Mahāyāna Buddhism, there was competition, too. Sri Lankan *Encyclopaedia of Buddhism* was an international cooperative project, with several regional committees set up in other countries, the largest one being the Japanese committee. They even depute Japanese scholars for full-time work in the Encyclopaedia office in Sri Lanka to act as liaison and also help the revision of the translations. Kyomasa Hayashima 早島鏡正, Hidetomo Kanaoka 金岡秀友, Kosuke Tamura 田村晃祐, and Sodō Mori 森祖道 were sent to Sri Lanka in different periods (Malalasekera [1997] 1961–1965, p. iv; Sodō 1968, p. 378). This inevitably affects knowledge expression. For instance, de Jong pointed out that in the preliminary version, "Quotations from Chinese Buddhist texts are made sometimes according to Nanjio's catalogue."[7] Kenneth Ch'en noted that in the encyclopedia, the Romanization of Chinese characters was inconsistent. Some contributors adhered to the Wade–Giles system, others followed the French method, while some did not follow any established system at all (Ch'en 1962, p. 369). Kenneth Ch'en might not be aware that, with regard to the Romanization of Chinese characters, there were at least two system, the Chinese and Japanese systems. The Chinese characters in Japanese were pronounced in a different way from those in Chinese, which bothered the committee and provoked debates. The Japanese worried that if all Chinese characters were Romanized according to Chinese pronunciation, the Japanese works written in ancient Chinese, which accounted for a large part of Japanese Buddhist works, would have titles that would never be pronounced even by their authors. In view of these conditions, they eventually decided that the works of Chinese authors use the pronunciation of Chinese, while the works of Japanese authors use the pronunciation

of Japanese (Sodō 1968, p. 383). The competition between China and Japan has lasted for a long time and spread to the field of encyclopedias.[8] This represents a big advancement for the Chinese comparing to the situation in a previously published Buddhist encyclopedia *Hōbōgirin*, in which all the Romanization of Chinese characters was undertaken according to Japanese pronunciation.

The Chinese editorial committee was cognizant of the complicated regional competition, and they tried to take the initiative to avoid this subordinated position in knowledge representation when they partook in the editing of the Sri Lankan Buddhist encyclopedia. A note about the progress of the ECB1956 project kept in the archives mentioned that "most of these entries Sri Lankan editorial board required are about esoteric Buddhism…if we don't provide, then they probably find some westerners to write them…therefore it is better for us to provide".[9]

In addition to the work on these details, the Chinese committee has made a greater effort to avoid a secondary and peripheral role in this knowledge production activity. The ECB1956 was initiated by Sri Lanka, and its primary purpose was to supply entries for the Sri Lankan *Encyclopaedia of Buddhism*, but the Chinese committee did not limit itself to this primary purpose. As mentioned above, by 1966, the committee had generated a total of 357 English manuscripts for the Sri Lankan Buddhist encyclopedia, only 154 of which had been sent to Sri Lanka (Fan 2018, p. 130), which reveals that they were not merely working for the Sri Lankan encyclopedia but had their own plans. In fact, they were compiling a complete encyclopedia for Chinese Buddhism. This can be seen from the categories of the manuscripts: history of Chinese Buddhism; Buddhist communication between China and neighboring countries; Buddhist sects in China, Buddhist figures in China, Buddhist rituals and regulations in China, Buddhist texts in China, Buddhism and Chinese culture; *Tripiṭaka* in China, etc.[10] The entries were organized by topics, each of which was arranged by timeline, which differed from the alphabetical order used in Sri Lankan encyclopedia. This can be further proven by the fact that the Chinese manuscripts were published later under the project of ZF1979.

The project of ZDQZ1980 was conducted entirely in Chinese, and the knowledge representation distinctly reflected Chinese local characteristics while also trying to maintain a global perspective. Yu Guangyuan 于光遠, the deputy director of the editorial committee of *Zhongguo daibaike quanshu*, emphasized while introducing the design of the encyclopedia that, "Our encyclopedia should not be regional. We should not degrade it to a regional encyclopedia… Our vision should not be limited by the region of China. The cultural knowledge of human beings, significant historical events, and scientific advancements all around the world—all things of importance—should be reflected in our encyclopedia…Science has no national boundaries."[11] Nonetheless, he also advocated that "Chinese characteristics must be underscored… Chinese history, Chinese geography, Chinese culture, and Chinese figures should all occupy more space, while less important things in foreign countries could be abridged. Only in this way can we express the 'of China' characteristics".[12] With respect to the Buddhist entries, the compilers, during the process, consciously referred to and gleaned insights from global encyclopedias, aspiring for their entries to meet the standards of world encyclopedias. For instance, the publishing house of *Zhongguo dabaike quanshu* circulated a journal, *Baike quanshu xueke cankao ziliao* 百科全書學科參考資料 (Reference for Subjects of Encyclopedia). This compiled data from various global encyclopedias and published sample entries of different subjects to serve as a guide for the authors and editors of the encyclopedia. In its special issue on religion, published in February 1980, entries such as "Buddhism", "Śākyamuni", and "Amitabha Sūtra" from *Gendai sekai hyakka daijiten* 現代世界百科大事典 (Encyclopedia of Modern World) and *Sekai daihyakka jiten* 世界大百科事典 (Grand Encyclopedia of World) of Japan, France's *Larousse Encyclopédique*, America's *Encyclopedia Americana*, and the United Kingdom's *Encyclopedia Britannica*, were selected and translated for reference by the domestic editorial board (Encyclopedia of China Publishing House 1980, pp. 1–34). On the other hand, the encyclopedia demonstrated a strong emphasis on Chinese content. For instance, within

the *Zhongguo daibaike quanshu: Zongjiao*, Buddhist entries accounted for 39.5%, Christianity 19.2%, and Islam 16.5%. Additionally, three indigenous Chinese branches were mentioned: Daoism (17%), religions of ethnic minorities in China (2.8%), and folk religions in China (0.7%) (See Figure 2). This highlights the compilers' intent to underscore Chinese information while maintaining a global viewpoint (Huang 1994, p. 247).

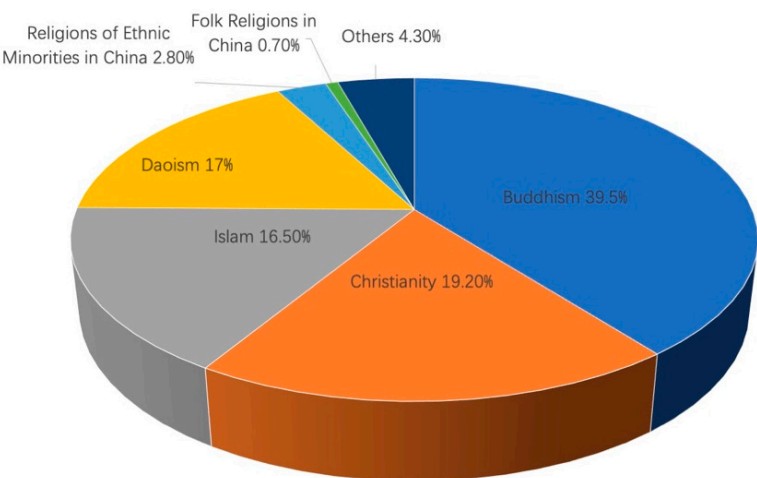

**Figure 2.** Entries of different religions in *Zhongguo dabaike quanshu: Zongjiao.*

Within the Buddhist entries, we can also see this balancing act between China and foreign countries, with a subtle preference for China. Taking the entries of figures as example, the *Zhongguo daibaike quanshu*: *Zongjiao* introduced 60 foreign Buddhist figures (51 ancient and 9 modern) and 84 Chinese figures (61 ancient and 23 modern) (Huang 1994, p. 249). This figure distribution reflects an evaluation of Buddhist figures from a Chinese standpoint, prioritizing those who profoundly influenced Chinese Buddhism.

In summary, when creating entries about Chinese Buddhism for the Sri Lankan Buddhist encyclopedia, the Chinese had to adapt to the local sensibilities of the Sri Lankan editorial board; when drafting entries for the *Zhongguo daibaike quanshu: Zongjiao*, the Chinese editors consciously highlighted Chinese characteristics and emphasized China's local aspects. This effectively illustrates that knowledge production is influenced by regional characteristics. It is not a wholly standardized and objective process, but fluctuates due to the variance in its place of origin. Knowledge expression is inevitably confined by the producer's capability, perspective, and standpoint.

## 4. Religious Knowledge in National Knowledge Systems

The formation of human knowledge systems was initially an unconscious undertaking. It was an endeavor to navigate the world more effectively, which led people to gather, systematize, and summarize information about their environment. Consequently, the first efforts to collect and organize knowledge were not undertaken by specialized scholars, but by individuals directly interacting with the world, such as farmers, herdsmen, fishermen, and explorers. Through confronting challenges and synthesizing experiences, they generated substantial first-hand knowledge for humanity. Coinciding with the evolution of human society, a specialized intellectual class gradually materialized, whose role was to elucidate the world to the masses, based on the knowledge amassed throughout history. In contemporary times, the breadth of human knowledge has become too vast, surpassing the comprehension of individuals or small groups. Therefore, large-scale knowledge organization endeavors and knowledge production projects are often state-led. Burke called this phenomenon the "nationalization of knowledge" (Burke 2012, p. 192). Helmholtz described that scholars came to be regarded as "representatives of their respective countries", recruited into "an organized army laboring on behalf of the whole nation" (von Helmholtz 1893, p. 24). For example, in ancient China, the Emperor Cheng of Ming

dynasty 明成祖 ordered Xie Jin 解縉 and Yao Guangxiao 姚廣孝 to preside over the compilation of *Yongle Dadian* 永樂大典 (Yongle Encyclopedia), a collection of ancient Chinese classics, comprising 22,937 manuscript rolls (卷) in 11,095 volumes (冊), with about 370 million words. Its ambition was to embody all the books since the beginning of Chinese civilization. Later, in the Qing dynasty, the Qianlong Emperor 乾隆帝 ordered the compilation of the *Siku Quanshu* 四庫全書 (Complete Library of the Four Treasuries), which was the largest collection of books in Chinese history, with 79,337 manuscript rolls, 36,381 volumes, 2.3 million pages and about 997 million words. With respects to the encyclopedias in this research, the Sri Lankan *Encyclopaedia of Buddhism* was championed by its Ministry of Culture Affairs, and the ECB1956 and ZF1979 were both spearheaded by the Chinese official organization BAC, while *Zhongguo dabaike quanshu* was a national-level project in China.

State involvement introduces new characteristics to knowledge production. For example, large-scale comprehensive encyclopedias have progressively become emblematic of national power and a platform for competition among nations. In the 19th century, a consensus was nearly unanimous among European countries that every civilized nation should possess its own encyclopedia as a testament to its prowess, thereby gaining respect from its neighbors and European peers (Kamusella 2009, p. 407). The compilation of *Enciclopedia italiana* is a good example for this. Italy started its encyclopedia compilation work in 1929, which is relatively late comparing to other European countries such as France, England, Germany and even Spain. The *Enciclopedia italiana* carried the mission of promoting all things Italian. The entry for "Garibaldi" takes up 17 columns, as compared to no more than 1 column for the corresponding entry in *Brockhaus* and *Larousse*. "Milan" continues for 59 columns, while there are only 7 columns in *Brockhaus* and *Larousse* (Burke 2012, p. 196). The compilation of encyclopedias in China also mirrored this sentiment. Examining the underlying motivation for compiling *Zhongguo dabaike quanshu*, the spirit of national competition was palpable: Jiang Chunfang said at the beginning of his "Recommendations for Editing and Publishing the Encyclopedia of China" that this is a "historical task and objective necessity" and that "…major countries worldwide have published extensive multi-volume encyclopedias since the mid-18th century… Presently, third-world countries are sequentially publishing encyclopedias, and even small countries like Suriname, which has not long gained independence, are also compiling and printing their own".[13] He dedicated approximately half of the article to recounting the history of encyclopedia compilation across various countries, thereby summarizing the contemporary trend of encyclopedia compilation globally.

The rationale behind states, as administrative entities, investing considerable financial and material resources in knowledge production is the intrinsic linkage between knowledge and power. The creation of knowledge systems often occurs concomitantly with the formulation of ideology. For instance, along with the compilation of the *Siku Quanshu*, the Qing Empire oversaw a catastrophe by banning and destroying books. They divided the classics into three levels: compiling, preserving the catalogue, banning and destroying. Any books considered by the rulers of the Qing Dynasty to violate Confucian ethical principles, ridicule the ancestors of the Manchus, or endanger the ruling status of the imperial family would be banned and destroyed. In 20 years, nearly 3000 kinds of books were destroyed (Li and Ju 2001, p. 48). With respect to the cases in my research, when the Sri Lankan encyclopedia was initiated, Sri Lanka had just concluded a colonial period that spanned over a century. During these extended colonial years, traditional Sri Lankan religions, including Buddhism, were persecuted, while Christianity, a representative of Western culture, was advocated. The compilation of the *Encyclopaedia of Buddhism* following the colonial era carried profound implications, such as the revitalization of local culture and the reestablishment of traditional beliefs.

In the context of compiling practices, the Sri Lankan *Encyclopaedia of Buddhism* holds significant official backing. Initially, the Buddhist Council of Sri Lanka (Laṅkā Bauddha Maṇḍalaya) was responsible for the encyclopedia's compilation, which was later handed over to the Ministry of Culture for overseeing the rest of the process. Malalasekera, the

initiator of this encyclopedic endeavor, was a multifaceted figure—a scholar, a Sri Lankan politician, and a social activist. A year after the encyclopedia's inception, he departed from the University of Sri Lanka, embarking on a diplomatic career. Subsequently, he held positions as an ambassador to the League of Nations and the High Commissioner to the UK. After a decade abroad, he returned to his homeland in late 1966. He resumed his work, serving as the chairman of the National Council for Higher Education, and was even considered for the role of the new governor-general (Sodō 1968, pp. 379–80).

In terms of expressing specific knowledge, the Sri Lankan editorial board made an earnest effort to amplify their own culture, while distancing it from Western influences. For instance, in the Sri Lankan *Encyclopaedia of Buddhism*, the Anno Domini dating system was used, but the term A.C. replaced A.D. when referring to the Common Era. This was explicitly mentioned in the book's preface: "In the text, as published in this volume, B.C. given with a date means 'Before Christ,' and A.C., 'After Christ'." (Malalasekera 1957, p. ix). This notation involved the relationship between religious influences and knowledge expression, as pointed out by Soper in his commentary on the book: "Chronology sensibly depends on the Christian era, with the small reservation implied by the use of A. C. rather than A. D." (Soper 1963, p. 366).

The compilation of the encyclopedia took place in the mid-twentieth century, during which the Anno Domini system—devised by the Christian theologian Dionysius Exiguus in the fifth century—was globally prevalent. This system uses the birth year of Jesus Christ as the inaugural year of the era, with the common abbreviations being B.C./A.D. In modern society, in an effort to maintain political correctness and religious neutrality, CE/BCE is often used instead, signifying Common Era/Before Common Era. While there are less common abbreviations such as A.A.C. (Anno ante Christum) and A.C. (Ante Christum) indicating "before Christ", the use of A.C. as "After Christ" is generally avoided in English, given that A.C. is conventionally understood to mean "before Christ" (Ante Christum).

To my knowledge, neither Malalasekera nor the editorial board provided explicit reasoning behind their preference for A.C. over A.D. Nonetheless, considering the circumstances, it may be attributed to their stance towards Christian culture. The term "Domini" in A.D. signifies God and carries strong Christian undertones, whereas the terms "Christ" in A.C. and "B.C." refer to the historical figure of Jesus Christ, bearing less religious weight.

The widespread adoption of the Anno Domini system mirrors the global expansion of Western civilization, intrinsically tied to Christianity. This propagation met varying degrees of resistance from local groups across different regions. The compilation of the encyclopedia occurred shortly after Sri Lanka emerged from its colonial period, during which it sought to diminish Western colonial influences and revitalize traditional Sinhala culture, with Buddhism at its center. Yet, in a project honoring the 2500th anniversary of Buddha's birth, it was somewhat awkward to employ the Anno Domini system that was globally accepted at the time. This episode underscores how international cultural dynamics can influence the nuanced articulation of knowledge.

As knowledge becomes nationalized, the ideology of a country profoundly influences its representation. For instance, the placement of specific types of knowledge within the overall knowledge system—as pointed out by Burke, in a knowledge map, different subjects occupy varying positions from the center to the periphery (Burke 2012, p. 198)—is intimately tied to a country's cultural backdrop and prevailing ideological trends. For example, in Sri Lanka, a country marked by its religious character, Buddhism is interwoven with the national convictions of its predominant ethnic group, the Sinhalese. As such, Buddhism occupies the central role in Sri Lanka's cultural traditions, with the *Encyclopaedia of Buddhism* serving as a critical component of the nation's core knowledge system and holding a close relationship with local traditional culture and values. China's situation, however, differs. Among the three Buddhist encyclopedia compilations mentioned earlier, the first two were curated by the BAC, with Buddhist knowledge undeniably forming the compilation's core. Yet, the assembling of Buddhist entries in the *Zhongguo dabaike quanshu: Zongjiao* diverges from these prior two compilations. Since the early 20th century,

China has been undergoing a transformation fueled by the idea of "science", resulting in religious beliefs being occasionally deemed as "superstitious", contrasting with "scientific" knowledge encompassing history, physics, and the like.

In 1981, an internal reference document titled *Zhongguo dabaike quanshu: Zongjiao* juan bianxie yaodian 《中國大百科全書·宗教》卷編寫要點 (Key Points of Compiling the *Zhongguo dabaike quanshu: Zongjiao*) stated the writing requirements as follows: "In crafting the entries for the Religion volume, we must maintain a strong focus on their scientific nature… we ought to neither propagate nor assail religion; we should strive to elucidate the content of the classics without offending religious believers or serving as propaganda to non-religious individuals. Materials must be carefully selected with a scientific mindset, and the foundational knowledge of religion should be conveyed to readers objectively, systematically, generally, and factually, portraying the true course of history."[14] With regards to the design of Buddhist entries for this encyclopedia, in "*Zhongguo dabaike quanshu: Zongjiaoxue* juan kuangjia, tiaomu caoan (Zhengqiu yijian gao)" 《中國大百科全書·宗教學》卷框架, 條目草案（徵求意見稿） (The Framework and Contents of *Zhongguo dabaike quanshu: Zongjiaoxue*—A Draft Seeking for Advices), dated to August 1979, a total of 420 Buddhist entries were proposed, subdivided into eight categories, including a general introduction; sects and organizations; buddhas, bodhisattvas, spirits and gods, heavens; doctrines and theories; scriptures and other texts; figures; monks and nuns, rituals, festivals; ruins, temples and pagodas.[15] However, the published *Zhongguo dabaike quanshu: Zongjiao* had expanded categorizations: Indian schools, sects, Chinese schools, Chinese sects, organizations, historical events, figures, scriptures, writings, miscellaneous works, foundational texts of Chinese sects, historical tales, catalogues of sūtras, category books, doctrines, terminology, institutions, rituals, the institutions about living buddhas, retreats, renowned mountains, temples and towers, culture, arts. The finalized book featured far more detailed categories, while the proposed topics tied to religious beliefs such as buddhas, bodhisattvas, spirits and gods, and heavens were omitted. As Burke puts it, "Encyclopaedias and their categories may be viewed as expressions or embodiments of a view of knowledge and indeed a view of the world." (Burke 2012, p. 94). It is obvious that throughout the *Zhongguo dabaike quanshu: Zongjiao*'s compilation, greater emphasis was placed on "objective" content such as Buddhist history, scriptures and writings, doctrines and teachings, organizations and institutions, and social influence, while "subjective" beliefs such as buddhas and bodhisattvas received less attention. This reflects how, across different countries and cultural traditions, knowledge production embodies an implicitly stated ideological background, and knowledge producers consciously or unconsciously adhere to this ideological paradigm during the knowledge creation process. In 1980s China, a prevalent perspective was to steer clear of "propagandizing" or "attacking" religion as much as possible, and underscoring the "scientific nature" of narrative in the articulation of religious knowledge.

While Buddhist knowledge formed the core in the Sri Lankan encyclopedia, as well as in ECB1956 and ZF1979 in China, it was rendered a subsidiary of the national knowledge system in ZDQZ1980. In these disparate scenarios, the structure of knowledge presentation and the narrative style underwent changes. Within this transformation, one can discern the shift and flow of knowledge power. Concurrently, it becomes evident that regardless of the position religious knowledge occupies in the entirety of the Chinese knowledge system, owing to Buddhism's unique standing in Chinese culture, Buddhist knowledge undoubtedly takes precedence in the Chinese religious knowledge system. Meanwhile, the expression of knowledge mirrors the ideological trend of their time.

## 5. Variability and Invariability of Buddhist Knowledge in Cultural Inheritance

In addition to the influence of region, politics, cultural customs, etc., a knowledge system develops according to its own internal orders, too. For example, knowledge production is affected by the intellectual traditions of the knowledge workers themselves. Usually, knowledge is inherited via two main mediums, namely, the texts and the people

who produce the texts. In the context of this research, the texts refers to the encyclopedia manuscripts, and the people denotes the authors of the encyclopedia entries. The three encyclopedia compilations discussed herein spanned two to three decades and involved two generations of scholars. Initial manuscripts underwent numerous revisions, with pertinent records maintained in archival repositories. Thus, we are able to observe both the variability and the invariability of Buddhist knowledge, as well as its producers, throughout the process of cultural inheritance.

The five-volume *Zhongguo Fojiao* produced in the ZF1979 project contains a total of 417 entries, distributed among 11 topics. Most of them were compiled and revised from the manuscripts that were composed in the ECB1956 project. Among the entries, topics such as the history of religion, the Buddhist relations between China and neighboring countries, sects, figures, ritual systems, and scriptures primarily adhered to those seen in the original manuscripts, with a small number of revisions.

The Buddhist entries of *Zhongguo dabaike quanshu: Zongjiao,* compiled in the ZDQZ1980 project, were not exclusively centered on Chinese Buddhism, but sought to encapsulate a comprehensive panorama of Buddhism. Hence, of the more than 400 entries, a substantial fraction focused on foreign Buddhism, most notably Indian Buddhism and Japanese Buddhism. This is a distinguishing feature compared to the preceding two compilations.

Nevertheless, a close interrelationship between the ZDQZ1980 and ZF1979 projects is discernible. For instance, they both incorporated classifications of sects, figures, *Tripiṭaka* versions, scriptures, teachings, institutions, and pagodas. Additionally, they contained entries bearing identical titles, with some even penned by the same author (see Table 2). For instance, the two entries on "Huijiao 慧皎" in *Zhongguo Fojiao* and *Zhongguo dabaike quanshu: Zongjiao* were both authored by Su Jinren 蘇晉仁. The core content and the emphasized information in the two entries were strikingly similar, with the only difference being that the entry in *Zhongguo dabaike quanshu: Zongjiao* was more succinct and compact, conforming to the writing requirements of the ZDQZ1980 project (Buddhist Association of China 1982, p. 85; Compilation Committee of Religion in the General Compilation Committee of *Zhongguo dabaike quanshu* 1988, p. 167).

The existence of an inheritance relationship between the two encyclopedias can be ascribed to two primary reasons. Initially, knowledge, as the vehicle for information, records historical facts, which remain constant; thus, knowledge archiving historical information remains invariant over time. Consequently, classics are usually unalterable. Secondly, the consistency of the authors fosters the consistency of knowledge expression: BAC participated in all three compilations, and numerous authors contributed to two or three of the aforementioned encyclopedia projects.

Comparing the authors of *Encyclopedia of Chinese Buddhism*, *Zhongguo Fojiao*, and *Zhongguo dabaike quanshu: Zongjiao*, it can be seen that:

a. Eight authors participated in three editing activities—Guo Yuanxing 郭元興, Huang Chanhua 黃懺華, Juzan 巨贊, Li An 李安, Lin Ziqing 林子青, Longlian 隆蓮, Tian Guanglie 田光烈 and You Xia 遊俠;

b. Four authors participated in the projects of ECB1956 and ZF1979—Gao Guanru 高觀如, Lü Cheng 呂澂, Yu Zhensheng 禹振聲 and Zhou Shujia 周叔迦;

c. Nine authors participated in the projects of ZF1979 and ZDQZ1980—Fang Guangchang 方廣錩, Liu Feng 劉峰, Ren Jie 任傑, Wang Sen 王森, Wang Xin 王新, Yao Changshou 姚長壽, Zhang Zhongxing 張中行, Zhou Shaoliang 周紹良 and Zhao Puchu 趙樸初[16];

d. Five authors participated in the projects of ECB1956 and ZDQZ1980—Fazun 法尊, Li Rongxi 李榮熙, Su Jinren 蘇晉仁, Yu Yu 虞愚 and Zhang Keqiang 張克強 (Zhang Jianmu 張建木).

**Table 2.** Some entries with the same authors in ZF1979 and ZDQZ1980.

| Author | *Zhongguo Fojiao* | *Zhongguo Dabaike Quanshu-Zongjiao* |
|---|---|---|
| Guo Yuanxing 郭元興 | Bukong 不空, Padmasambhava 蓮華生, Great Perfection 大圓滿 | Bukong 不空, Padmasambhava 蓮華生, Great Perfection 大圓滿 |
| Fazun 法尊 | Bkaḥ-Gdams-Pa Sect 迦當派 | Bkaḥ-Gdams-Pa Sect 噶當派 |
| Huang Chanhua 黃懺華 | Masters of the Abhidharmakośa-Śāstra 俱舍師 | School of the Abhidharmakośa-Śāstra 俱舍學派 |
| Juzan 巨贊 | Dao'an 道安, Yixing 一行, Yanshou 延壽 | Dao'an 道安, Yixing 一行, Yanshou 延壽 |
| Lin Ziqing 林子青 | Masters of the Satyasiddhi-Śāstra 成實師, Masters of the Parinirvāṇa-Sūtra 涅槃師, Hongyi 弘一, Jing'an 敬安, Three-Stage Teaching 三階教, Yinguang 印光 | School of the Satyasiddhi-Śāstra 成實學派, Masters of the Parinirvāṇa-Sūtra 涅槃學派, Hongyi 弘一, Jing'an 敬安, Three-Stage Teaching 三階教, Yinguang 印光 |
| Longlian 隆蓮 | Kuiji 窺基 | Kuiji 窺基 |
| Su Jinren 蘇晉仁 | Huijiao 慧皎 | Huijiao 慧皎 |
| Tian Guanglie 田光烈 | Masters of the Daśabhūmi-Vyākhyāna 地論師, Masters of the Mahāyānasaṃgraha-Śāstra 攝論師 | School of the Daśabhūmi-Vyākhyāna 地論學派, Masters of the Mahāyānasaṃgraha-Śāstra 攝論學派 |
| You Xia 遊俠 | Masters of the Abhidharma 毗曇師 | School of the Abhidharma 毗曇學派 |
| Yu Yu 虞愚 | Ci'en School 慈恩宗 | Faxiang School 法相宗 |
| Zhang Jianmu 張建木 | Zhidun 支遁, Tsoṅ-kha-pa 宗喀巴 | Zhidun 支遁, Tsoṅ-kha-pa 宗喀巴 |
| Zhou Shaoliang 周紹良 | Popular Preaching and Sūtra Preaching 俗講和講經文 | Sūtra Preaching 講經文 |

During the 1980s, when the ZF1979 and ZDQZ1980 projects were underway, numerous scholars who had participated in the ECB1956 project in the 1950s and 1960s were approaching the later years of their lives, including Lü Cheng (1896–1989), Li An (1900–1985), and Juzan (1908–1984), while some had passed away, such as Xirao Jiacuo (1883–1968), Zhou Shujia (1899–1970), Fazun (1902–1980), and Gao Guanru (1906–1979). Meanwhile, a fresh cohort of Buddhist scholars emerged, becoming the new driving force of Buddhist academia, including individuals such as Fang Guangchang, Liu Feng, Wang Xin, and Yao Changshou. Additionally, when *Zhongguo dabaike quanshu: Zongjiao* was eventually published in 1988, entries on figures such as "Fazun", "Juzan", "Lü Cheng", "Xirao Jiacuo" and "Zhao Puchu", etc. (Compilation Committee of Religion in the General Compilation Committee of *Zhongguo dabaike quanshu* 1988, pp. 108, 218, 245, 429, 518)—the creators of the Chinese Buddhist knowledge system from the 1950s and 1960s—were all encapsulated in the new encyclopedia. In this way, those who shaped history became a part of that history. Through the continued work of two generations of scholars, the knowledge system of Buddhist encyclopedias has also been inherited and further cultivated. Using the topic of *Tripiṭaka* as an example, during the compilation of ECB1956, entries introducing China's *Tripiṭakas* were primarily authored by Lü Cheng, a widely renowned specialist in the field during that time. However, in the 1980s and 1990s, as the fifth volume of *Zhongguo Fojiao* was being compiled, due to advancements in archeological work and academic

research, new discoveries regarding *Tripiṭaka* were made in contemporary academia. As a result, the information within the earlier manuscripts appeared somewhat antiquated. Therefore, the editorial team of *Zhongguo Fojiao* invited Fang Guangchang, an expert on Buddhist scriptures, to amend the prior entry and pen a few new entries grounded on the most recent research, thereby presenting the most recent and comprehensive information concerning *Tripiṭaka*. Furthermore, entries pertaining to the grottoes included in the fifth volume of *Zhongguo Fojiao* represented the combined efforts of two generations of scholars over a span of two to three decades. Initially composed by Yan Wenru 閻文儒 in the 1960s, they were later supplemented and updated by his student Qi Qingguo 祁慶國 in the 1980s (Buddhist Association of China 2004, p. 533).

In summary, throughout the three compilations, Buddhist knowledge has been handed down both through texts and scholars. Given the consistency of historical facts and the stability of knowledge producers, the outcomes of the three compilations exhibited numerous similarities. Nevertheless, Buddhist knowledge has also evolved and undergone some modifications in the process of being transmitted from one generation to the next. The inheritance and modification of knowledge in this historical progression have left imprints in the working archives and final results of the three compilations.

## 6. Conclusions

The sociology of knowledge argues that knowledge is a construct, with its formation shaped by myriad social factors such as regional culture, ideological currents, international relations, and the positionality of knowledge producers. The knowledge system delineated by an encyclopedia serves to construct societal common sense, impacting the intellectual life of the populace. It is this crucial function that underscores the importance for a nation or cultural tradition of establishing its own knowledge system through the compilation of encyclopedias. This paper explores the undertaking of constructing the Chinese Buddhist knowledge system, revealing the efforts of the Chinese Buddhist community and the implicit or explicit influences of varied societal aspects through the examination of three Buddhist encyclopedia compilations, as well as the simple description of a few other encyclopedias.

Buddhism, while originating from foreign lands, has thrived in China, evolving into a pivotal component of Chinese culture, alongside the indigenous traditions of Confucianism and Daoism. Consequently, Buddhist knowledge naturally embodies both global and local attributes. The compilation process of the Sri Lankan *Encyclopaedia of Buddhism* and the Chinese *Zhongguo dabaike quanshu: Zongjiao* present a common scenario: the editorial team strived to create a globally inclusive encyclopedia, but inadvertently or consciously, the final product mirrored a local hue. This regional idiosyncrasy in knowledge is partially attributed to the constraints of the editorial board, and partially due to the ideological directions of a nation, considering that encyclopedia compilations are typically substantial national projects. The influence of national ideological tendencies on the articulation of knowledge is substantial. Across nations, religious knowledge occupies varying degrees of prominence within the broader knowledge system, and within that, different religions secure different positions. For instance, Buddhist knowledge assumes a central role in Sri Lanka, a country where Buddhism is the state religion. However, the scenario in 1980s China was quite different: historical reasons placed religious knowledge at a relatively peripheral position within the national knowledge system, yet Buddhism received emphasis due to its status as the most Sinicized among the world's three major religions. It is precisely these inherently global and local characteristics of Buddhism, and the unique and crucial position that Buddhist knowledge holds within the Chinese religious knowledge system, that prompted three official compilations of Buddhist encyclopedias within a brief timespan. Through this lens, we can trace the lineage and transformation of knowledge, as well as the enduring aspects within historical continuity.

In addition to the regional, political and cultural elements, intellectual traditions also affect the expression of knowledge. As Lemaine puts it, the institutional context of knowledge plays a very important role in knowledge history (Lemaine et al. 1976, pp. 8–9); for example, a stable institutional environment would result in stable knowledge expression. Quite a few authors participated in two or three compilations, which results in the inherited nature of these encyclopedias. It is noteworthy that Chinese Buddhism has a tradition of compiling encyclopedic works. In their history, Buddhists have compiled many books of this kind, such as *Fayuan zhulin* 法苑珠林 (The Pearl Grove of the Garden of the Dharma), *Dacheng yizhang* 大乘義章 (Dictionary of Mahāyāna Buddhism) and *Zongjing lu* 宗鏡錄 (A Record of the Mirror of the Tenet of the Chan School), all of which reveals the overall picture of Chinese Buddhism. Therefore, we can consider the three compilations of encyclopedias for Chinese Buddhism as a continuation of this tradition.

**Funding:** This research was funded by the Youth Project of 2020 National Social Science Fund of China, "The History of the Evolution of Chinese Buddhist Knowledge System" (20CZJ006) and Major Project of 2017 National Social Science Fund of China, "Social Life History of Chinese Buddhist Monks" (17ZDA233).

**Institutional Review Board Statement:** Not applicable.

**Informed Consent Statement:** Not applicable.

**Data Availability Statement:** No new data were generated or analyzed in support of this research.

**Conflicts of Interest:** I declare there is no conflict of interest.

## Notes

[1] BAC was responsible for the compilation of entries about Chinese Buddhism in the Sri Lankan encyclopedia.

[2] They are the editors-in-chief for the Buddhist part of *Zhongguo dabaike quanshu: Zongjiao* 中國大百科全書: 宗教 (Encyclopedia of China—Religion).

[3] This is the number of participants in the project, including the authors, translators and organizers.

[4] 21 authors engaged in the publishing project of *Zhongguo fojiao* after 1980. Therefore, with the 38 participants of ECB1956 added, 59 is the overall number for this project.

[5] This is the number of authors according to the author list.

[6] To my knowledge, BAC has been working for the publication of this encyclopedia since 2018. It is forthcoming in Sino-Culture Press (Huawen chubanshe 華文出版社) in China.

[7] Nanjio's catalogue refers to Bunyiu (1883).

[8] The library of Research Institute of Chinese Buddhist Culture, BAC preserved a recommendation letter for Youdao 由道, a translator of ECB1956, in which Zhao Puchu mentioned that the Editor-in-Chief of the Sri Lanka Encyclopedia used to tell him that, "Chinese manuscripts are far more better than Japanese ones." (Zhao 1992).

[9] …錫方要求撰寫的這些文稿, 多係密教條目…我們如不供應, 則他們很可能找些西方人寫…因此仍以我們供應為宜. This document is preserved in manuscript archives of the *Encyclopedia of Chinese Buddhism*. Fragments 1—proofread notes and letters—letter 4—00002.

[10] This classification was inherited by later Chinese Buddhist encyclopedias, for example, in *Zhongguo fojiaobaike quanshu* 中國佛教百科全書 (Encyclopedia of Chinese Buddhism), an eight volumed work published by Lai (2000), the 3,000,000-word manuscript was classified into the following categories: classics, doctrines, figures, history, Buddhist schools, rituals and regulations, poems and *gāthās*, calligraphies and paintings, sculptures, architectures, renowned mountains and monasteries.

[11] 我們這部大百科全書, 不應該是地區性的, 不能自己把這部大百科全書降到一個地區性的水平…我們的視野不要受中國這個地區的限制, 對全世界人類的文化知識, 全世界歷史上發生過的大事, 全世界科學的發展, 凡是重要的, 都應該在我們這部大百科全書中得到反映…科學是沒有國界的. See (Yu 1980, p. 19).

[12] 中國的特點是必需強調的…中國歷史, 中國地理, 中國的文化, 中國的人物都應該有較多的篇幅, 外國不那麼重要的東西可以簡略些. 這樣才可以寫出"中國的'"這個特點. See (Yu 1980, p. 20).

[13] …世界各主要國家, 從18世紀中葉開始就出版大型的多卷本的百科全書…現在第三世界國家, 也紛紛出版百科全書, 連獨立不久的蘇里南這樣的小國, 也在編印. See (Jiang 1990, p. 1).

14 撰寫《宗教》卷條目釋文, 一方面要十分注意科學性…應堅持既不宣揚宗教, 亦不攻擊宗教; 力求釋文內容既不傷害宗教信仰者的宗教感情, 又不成為向不信仰宗教的群眾進行傳教的材料. 應以科學態度精審地選取材料, 客觀地, 系統地, 概括地, 實事求是地向讀者介紹宗教的基本知識, 還事物的歷史本來面目. This document is preserved in manuscript archives of the *Encyclopedia of Chinese Buddhism*. Refrences—*Zhongguo dabaike quanshu: Zongjiaojuan*—Key points—00002.

15 This document is preserved in the manuscript archives of the *Encyclopedia of Chinese Buddhism*. Refrences—*Dabaike quanshu: Zongjiaoxue*—A draft seeking for advice—00011.

16 Zhao Puchu engaged in the ECB1956 project as the project leader, but not as an author.

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
