# Peer review of "Regional, Ideological and Inheritable Characteristics of Knowledge: A Survey of Three Compilations of Buddhist Encyclopedias in China from 1950s to 2000s"

_religions, doi:10.3390/rel14111336_

Round 1

Reviewer 1 Report

This article presents an interesting case of the religious knowledge production and transmission, using comparison of three modern Chinese Buddhist encyclopedias. The author has a very good command of the history of their production, the article is written in clear, easy comprehensible and logical style. The manuscript can be improved though, mainly in terms of adding a broader perspective on this subject, so that it can appeal to broader audience in this field.

Several basic suggestions:

1.      Start introduction with the direct presentation of goals and research methods, more detailed information on sources can be moved to the later paragraph.

2.      Strengthen the conclusion with the broader theoretical and comparative views.

3.      Include some theoretical foundations of the “knowledge production”, construction of national knowledge systems, especially as concerns religious compendia (encyclopedias).

4.      More comparison with the similar projects in other countries (Western, Japanese, and Korean) will be very useful, as well as the diachronical comparison with the later and earlier periods of the history of Chinese encyclopedias. This can clearer demonstrate the special features of the encyclopedias under study, which will be especially valued by non-specialist readers.

Reviewer 2 Report

This is a very thorough and detailed study of the three compilations of Buddhist encyclopadias in China from 1950s to 2000s. There is really no doubt that the author has done a good job in collecting data. My suggestion is mainly about the so-what question of this research. As the author pinpoints, such compilation shows a way of producing knowledge in the Buddhist context at that time. And the author also shows how knowledge production features in cultural exchange and ideology making. But I want to encourage the author to further the discussion: if this is the case, what is the power dynamics that underpin such knowledge production? As the author also specifies that knowledge is a human construct that started in history as an unconscious effort but gradually takes on various tasks. If knowledge production is not unconscious, how it contributes to and consolidates power inside and outside Buddhist communities? 

More importantly, knowledge makers, how did they exercise their agency? When account for the first compilation of Buddhist encyclopedia, the author revealed that members in the Chinese section knew about their marginalisation. Hence, they were very much mindful of their position but they still chose to join the editorial mission. Later on, the author also lists the intellectuals who participated in editing and compiling these encyclopaedias. What then is their role in the interplay between knowledge production and power? How shall we understand the agency of the compliers/workers along their compilation/work? 

Reviewer 3 Report

This article clearly discusses the background and process of compiling the three Buddhist encyclopedias between 1950s and 2000s, and it strikes a good balance between fact and opinion, with a touching account of the circumstances and limitations of writing for different generations of Buddhist scholars.

(1) I would suggest: 

Add a (pie) chart to the “Introduction” , visualize the information in this section, i.e., three encyclopedias’ compiling process, the sponsor, the participants, the content ( e.g. the 357 English manuscripts, 154 of which sent to Sri Lanka, while the Chinese manuscripts published in ZF1979, etc.). This will help to clarify the links, overlaps and differences between the three processes, and elucidate vividly the conditions, constraints and mechanisms of knowledge production. 

(2) The article’s application of power-knowledge paradigm is inspiring, meanwhile oversimplified. For example, with the rise of modern Sri Lanka, the compilation of large-scale comprehensive encyclopedias has taken off their way, furthermore, “large-scale comprehensive encyclopedias have progressively become emblematic of national power and a platform for competition among nations ” (272-273), but this power-knowledge linkage could not convincingly explain why ZDQZ1980 “sought to encapsulate the comprehensive panorama of Buddhism”(419). 

To make the argument clearer, I would suggest to separate the core and out-layer of knowledge production process. The former involves the “products” and the “producers”, e.g. the existence of an inheritance relationship between the two encyclopedias (ZF1979 and ZDQZ1980, 437), numerous authors had contributed to two or three of the aforementioned encyclopedia projects (442), etc.

The out-layer of knowledge production, means geographical characteristics, institutional differences, state ideology, or, context and constraints of compilation of encyclopedia. 

(3) One dimension that the author should be aware of is, China has its own tradition of large-scale compilation of encyclopedia or 类书. From this point of view, the modern efforts of ECB1956, ZF1979 and ZDQZ1980 are not only to show national power, but new forms of old ideals to certain degree . This knowledge-construction basis will support the Chinese Buddhist scholars empower themselves even under the constraints of state ideologies.

Reviewer 4 Report

A sociological examination is meritorious. However, the time period from the 1950s to the 2000s is quite long, and the society has changed tremendously as a result of China's dynamic political and economic contexts. Given this premise, it might be better to demonstrate the diverse social, political, and economic influences on ideological tendencies in your paper. 

For me, English language of the paper is quite okay. 
